

# Characterization of the ATPase FlaI of the motor complex of the *Pyrococcus furiosus* archaellum and its interactions between the ATP-binding protein FlaH

Paushali Chaudhury, Chris van der Does and Sonja-Verena Albers

Molecular Biology of Archaea, Institute of Biology, University of Freiburg, Freiburg, Germany

## ABSTRACT

The archaellum, the rotating motility structure of archaea, is best studied in the crenarchaeon *Sulfolobus acidocaldarius*. To better understand how assembly and rotation of this structure is driven, two ATP-binding proteins, FlaI and FlaH of the motor complex of the archaellum of the euryarchaeon *Pyrococcus furiosus*, were overexpressed, purified and studied. Contrary to the FlaI ATPase of *S. acidocaldarius*, which only forms a hexamer after binding of nucleotides, FlaI of *P. furiosus* formed a hexamer in a nucleotide independent manner. In this hexamer only 2 of the ATP binding sites were available for binding of the fluorescent ATP-analog MANT-ATP, suggesting a twofold symmetry in the hexamer. *P. furiosus* FlaI showed a 250-fold higher ATPase activity than *S. acidocaldarius* FlaI. Interaction studies between the isolated N- and C-terminal domains of FlaI showed interactions between the N- and C-terminal domains and strong interactions between the N-terminal domains not previously observed for ATPases involved in archaellum assembly. These interactions played a role in oligomerization and activity, suggesting a conformational state of the hexamer not observed before. Further interaction studies show that the C-terminal domain of *Pf* FlaI interacts with the nucleotide binding protein FlaH. This interaction stimulates the ATPase activity of FlaI optimally at a 1:1 stoichiometry, suggesting that hexameric *Pf* FlaI interacts with hexameric *Pf* FlaH. These data help to further understand the complex interactions that are required to energize the archaellar motor.

## INTRODUCTION

Motility in archaea is driven by a rotating cell surface appendage, called the archaellum (*Jarrell & Albers, 2012*; *Albers & Jarrell, 2015*). The archaellum is widely spread in archaea, and was identified in many of the archaeal phyla, e.g., in crenarchaeota, euryarchaeota, thaumarchaeota and nanoarchaeota (*Makarova, Koonin & Albers, 2016*). Although in function it resembles the bacterial flagellum, it is structurally different as it is evolutionarily related to archaeal and bacterial type IV pilus assembly systems (T4PSs) and Type II secretion systems (T2SS) (*Jarrell & Albers, 2012*; *Berry & Pelicic, 2015*; *Albers & Jarrell, 2015*). The archaellum consists of 7–13 proteins, which are all essential for the assembly and function (*Patenge et al., 2001*; *Thomas, Pawson & Jarrell, 2001*; *Chaban et al., 2007*;

Corresponding author
Sonja-Verena Albers,
sonja.albers@biologie.uni-freiburg.de

*Lassak et al., 2012*). Similar to the pilins of T4PSs, archaellins also possess N-terminal class III signal peptides, which are processed by a dedicated membrane bound aspartic acid peptidase (*Albers, Szabó & Driessen, 2003*; *Bardy & Jarrell, 2003*). After N-terminal cleavage, the mature archaellins are inserted into the growing archaellum filament. Recently, cryo EM structures of the euryarchaeotes *Methanospirillum hungatei* and *Pyrococcus furiosus* archaellum revealed that the archaellin monomer has two domains: an N-terminal domain, which forms a long hydrophobic α-helix, and a C-terminal domain with an eight-stranded anti-parallel β-barrel (*Poweleit et al., 2016*; *Daum et al., 2017*). The assembled archaellins showed different inter-subunit interactions than the assembled pilins of T4PSs (*Craig et al., 2006*; *Wang et al., 2017*) and the Iho670 adhesion filament of non-motile *Ignicoccus hospitalis* (*Braun et al., 2016*). Similar to the assembly of the pilus in T4PS and T2SSs (*Jakovljevic et al., 2008*; *Chiang et al., 2008*; *Yamagata & Tainer, 2007*), assembly of the archaellum is energized by ATP hydrolysis (*Thomas, Pawson & Jarrell, 2001*; *Reindl et al., 2013*). Operons encoding components of the archaellum contain two ATP binding proteins, FlaI and FlaH. FlaI of the crenarchaeon *Sulfolobus acidocaldarius* (*Sa*FlaI) forms an ATP-dependent hexamer and the nucleotide bound crystal structure showed a conserved C-terminal ATPase domain (CTD) which is connected via a flexible linker to the variable N-terminal domain (NTD) (*Ghosh et al., 2011*; *Reindl et al., 2013*). In the hexameric crystal structure, an intrinsic twofold symmetry results in three unique subunit conformations, and superimposition of unique NTDs and CTDs shows that the individual NTD and CTD structures are similar, with only small changes in the NTD. The intra-subunit interface is largest between the CTDs. Indeed, *Sa*FlaI lacking the NTD still exhibits 75% of the ATPase activity compared to the full-length FlaI (*Reindl et al., 2013*). In the crystal structures, the interaction between the NTD and CTD from one subunit is small compared to the interaction between one NTD and the neighboring subunit CTD (*Reindl et al., 2013*). Together with the hexameric structures of Type II secretion and Type IV pili systems (*Yamagata & Tainer, 2007*; *Misic, Satyshur & Forest, 2010*; *Lu et al., 2013*; *McCallum et al., 2017*) a model evolved in which successive rounds of ATP binding, ATP hydrolysis and ADP release in the three unique subunits result in conformational changes of the subunits. Based on the homology of FlaI and FlaJ with the T4PS proteins, it seems likely that FlaI interacts with FlaJ (*Chiang, Habash & Burrows, 2005*; *Takhar et al., 2013*; *Bischof et al., 2016*). Indeed, the flexible crown groove (residues 61–128) of the structure of *Sa*FlaI contains negatively charged amino acid patches which were proposed to interact with the positively charged cytoplasmic loops of *Sa*FlaJ (*Reindl et al., 2013*; *Banerjee et al., 2013*). A similar interaction was recently proposed between the PilB and PilC of *Geobacter metallireducens* (*McCallum et al., 2017*). Conformational changes in the membrane platform protein then might result in insertion or extrusion of the pilin or archaellin into the pilus/archaellum (*Chang et al., 2016*; *McCallum et al., 2017*). Indeed, the ATPase activity of *Myxococcus xanthus* PilB is stimulated by interaction with PilC (*Bischof et al., 2016*). Additional to its assembly, the archaellum also needs to rotate. How the switch between assembly and rotation occurs is currently unknown, but possibly FlaH, a second ATPase only identified in operons encoding the archaellum but not in operons encoding archaeal Type IV pili systems, is involved in this switch (*Chaudhury*

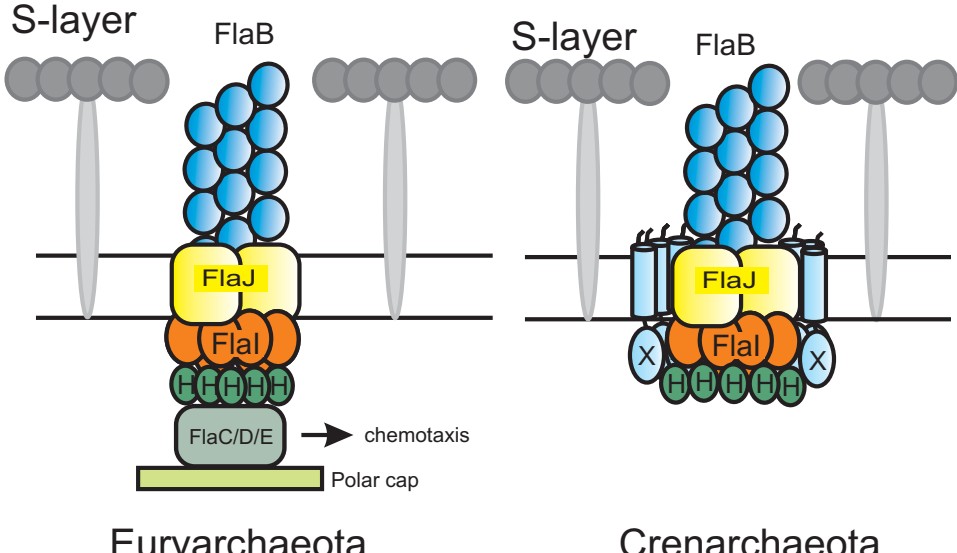

**Figure 1** **Current models of the euryarchaeal and crenarchaeal archaellum motor complex.** In both shown archaellum motor complexes, FlaH, FlaI and FlaJ are conserved. FlaB, the archaellin, builds the fiament in the archaeal cell envelope (S-layer). The euryarchaeal archaellum motor complex contains FlaC/D/E which are thought to interact with the chemotaxis system. Additionally, a polar cap structure was identified very recently. However, its function is so far unknown. In the crenarchaeal archaellum motor complex FlaX forms a ring-like structure and is thought to act as a scaffold protein for motor protein assembly.

_et al., 2016_). FlaH belongs to the RecA superfamily of ATPases. _S. acidocaldarius_ FlaH (_Sa_FlaH) can bind ATP, but is unable to hydrolyze it, most likely due to the presence of a non-canonical walker B motif. The crystal structures of _Sa_FlaH and _Methanocaldococcus jannaschii_ FlaH were solved (_Meshcheryakov & Wolf, 2016_; _Chaudhury et al., 2016_). _Sa_FlaI and _Sa_FlaH interact with each other in an ATP dependent manner (_Chaudhury et al., 2016_). _Sa_FlaI and _Sa_FlaH were also shown to interact with _S. acidocaldarius_ FlaX (_Sa_FlaX) (_Banerjee et al., 2013_). FlaX, which was only identified in crenarchaea, contains an N-terminal transmembrane domain and a C-terminal cytoplasmic domain which, for _Sa_FlaX, forms a ring-like oligomeric structure with a diameter of 30 nm (_Banerjee et al., 2012_). Deletion of 57 amino acids which correspond to three helices from the C-terminus of _Sa_FlaX abolished formation of the ring and interaction with FlaI _in vitro_ (_Banerjee et al., 2012_; _Banerjee et al., 2013_). Electron microscopy revealed that, _in vitro_, _Sa_FlaH could assemble as a second ring inside the _Sa_FlaX ring (_Chaudhury et al., 2016_). Thus, it was proposed that the central core of the crenarchaeal archaellum is formed by FlaI together with FlaH, FlaX and FlaJ (_Banerjee et al., 2013_) (Fig. 1). In addition to these proteins, a minimal functional archaellum requires the FlaF and FlaG proteins. FlaF and FlaG are monotopic membrane proteins where _Sa_FlaF has a β-sandwich fold and interacts with S-layer proteins suggesting that it might act as a stator that anchors the rotating archaellum (_Banerjee et al., 2015_).

Although the archaellum is widely spread among archaea, the archaellum has mainly been biochemically characterized in the crenarchaeote *S. acidocaldarius*. Several differences have been observed between the archaellum systems of crenarchaea and euryarchaea. For example, whereas most species of crenarchaeota contain only one archaellin, euryarchaeota may possess up to five different archaellins (*Jarrell & Albers, 2012*). In *Methanococcus maripaludis,* the archaellum showed a hook-like structure which was not observed in a deletion mutant of the minor archaellin *flaB3* (*Chaban et al., 2007*). Furthermore, FlaX was only identified in crenarchaea (*Ghosh & Albers, 2011*), whereas euryarchaeota contain the *flaC, flaD and flaE* genes which conversely are not found in crenarchaea (*Jarrell & Albers, 2012*; *Albers & Jarrell, 2015*) (Fig. 1). Finally, many euryarchaea exhibit chemotaxis systems, which have not been identified in crenarchaea till date (*Wuichet, Cantwell & Zhulin, 2010*). In *Halobacterium salinarum* and *Haloferax volcanii*, it has been demonstrated that the FlaC, FlaD and FlaE proteins link rotation of the archaellum to the CheY signal transduction cascade and thus to the chemotaxis system (*Schlesner et al., 2009*; *Quax et al., 2018*).

Recently, a low resolution image of the archaellar basal body of the euryarchaeote *Thermococcus kodakaraensis* was obtained by cryo-tomography (*Briegel et al., 2017*). This structure shows similarities to the structures of bacterial T4P (*Chang et al., 2016*), but also shows several unique features. For example, a large conical frustum of up to 500 nm in diameter was observed at the cytosolic base of the structure. The resolution of this structure is however not high enough to distinguish or identify individual components (*Briegel et al., 2017*). The cryo-EM structure of the *P. furiosus* archaellum allowed a more detailed view of the archaellum motor complex (*Daum et al., 2017*). As in *T. kodakaraensis*, a cone structure is present below the archaellum motor complexes. After modelling the structures of *Sa*FlaI and *Sa*FlaH in the densities close to the membrane, several remaining densities were observed. These probably contain the FlaCDE proteins.

In this study, we set out to characterize the motor subunits of the archaellum of *Pyrococcus furiosus,* an anaerobic, heterotrophic hyperthermophilic euryarchaeote that can grow at temperatures between 70 °C and 103 °C, and a pH between 5 and 9 (*Fiala & Stetter, 1986*). *P. furiosus* contains monopolar polytrichous archaella (*Fiala & Stetter, 1986*), which it does not only use to swim, but also to form cable-like cell–cell connections to adhere to solid surfaces (*Näther et al., 2006*). Here, we continue our previous studies on the biochemical characterization of *P. furiosus* FlaI (*Pf* FlaI) and FlaH (*Pf* FlaH).

## MATERIALS & METHODS

### Strains and plasmids

*Escherichia coli* strains NEB 10-beta (New England BioLabs) and Rosetta (DE3) (Novagen) were used for cloning purposes and overexpression respectively. Genomic DNA of *Pyrococcus furiosus* DSM 3638 (*Robb et al., 2001*) was used as a template for PCR reactions. Plasmids and their construction are described in Supplementary Table S1. Primers used are described in Supplementary Table S2. All plasmids sequences were confirmed by PCR and sequencing.

## Overproduction and purification

Overproduction and purification of His-tagged *P. furiosus* FlaI (*Pf* FlaI), *Pf* FlaI-NTD, *Pf* FlaI-CTD, and *P.furiosus* FlaH (*Pf* FlaH) and the *Pf* FlaH (K39A) and *Pf* FlaH (D126N) mutants were performed as described previously (*Chaudhury et al., 2016*). Overproduction and purification of StrepII-tagged *Pf* FlaI (E336A) were essentially performed the same except that Streptactin column material (IBA GmbH, Göttingen, Germany) was used, and that the protein was eluted with 2.5 mM d-desthiobiotin. Samples were stored at −80 °C until use.

## Analytical gel filtration

*Pf* FlaI, *Pf* FlaI-NTD and *Pf* FlaI-CTD were concentrated to 1 mg/ml in buffer containing 20mM Tris HCl pH 8.0, 150 mM NaCl (buffer A) using Amicon concentrators (Millipore) with a 10 kDa cut-off. 500 μl of the concentrated samples or 250 μl *Pf* FlaI-NTD mixed with 250 μl *Pf* FlaI-CTD and applied to Superdex 200 10/300 GL or Superdex 75 10/300 GL size exclusion columns equilibrated with buffer A. Fractions were analyzed on SDS-PAGE. Thyroglobulin (669 kDa), γ-globulin (158 kDa), ovalbumin (44 kDa), myoglobin (17 kDa) and vitamin $B_{12}$ (1.35 kDa) were used as size standards.

## MANT-ATP binding

Binding of the fluorescent ATP analog $2'$-/$3'$-$O$-($N$-methylanthraniloyl) adenosine $5'$-triphosphate (MANT-ATP, JENA biosciences) was detected by titrating *Pf* FlaI (20 nM, 100 nM and 5 μM in buffer A containing 5 mM $MgCl_2$) with increasing concentrations of MANT-ATP in a 150 μl cuvette at 20 °C in a Fluoromax-4 fluorimeter (HORIBA Scientific, Irvine, CA, USA). Excitation and emission wavelengths were set to 285 and 450 nm respectively with slit widths of 10 nm. Fluorescence was corrected for MANT-ATP fluorescence in the absence of protein. To determine the binding affinity of ATP, competition assays between MANT-ATP and ATP were performed. Total fluorescence was determined under the conditions described above after addition of increasing amounts of ATP to a solution containing 20 nM *Pf* FlaI and 10 nM of MANT-ATP. The data were fitted with the Hill equation: $F = (F_{\max} + (F_{\min} - F_{\max}) * [ATP]^n)/(IC_{50}^n + [ATP]^n)$, where $F =$ Fluorescence, $F_{\min} =$ minimal fluorescence, $F_{\max} =$ maximal fluorescence, [ATP] is the ATP concentration, $IC_{50}$ is the ATP concentration where the fluorescence is reduced by half, and $n =$ Hill coefficient.

## ATPase assay

Release of inorganic phosphate after ATP hydrolysis was determined using the Malachite green assay (*Lanzetta et al., 1979*) by determining the colorimetric change at 620 nm using a Clariostar plate reader (BMG labtech). ATP hydrolysis at different temperatures was determined by incubating 12.5 μg/ml *Pf* FlaI in buffer A containing 5 mM $MgCl_2$ and1 mM ATP for 5 min at different temperatures. ATP hydrolysis at different pHs was determined by incubating 12.5 μg/ml *Pf* FlaI in different buffers (20 mM citrate (pH 3.0), 20 mM 2-(N-morpholino)ethane sulfonic acid (MES, pH 6.0), 20 mM 4-(2-hydroxyethyl)-1-piperazine ethane sulfonic acid (HEPES, pH 7.2), 20 mM 2-amino-2-hydroxymethyl-propane-1,3-diol (TRIS, pH 8.0/ 9.5), 20 mM sodium carbonate-bicarbonate (pH 10.0)) containing 5 mM

MgCl$_2$ and1 mM ATP for 5 min at 70 °C. To determine the ATPase activity of $Pf$ FlaI at different ATP concentrations, 12.5 μg/ml $Pf$ FlaI was incubated for 5 min at 70 °C in buffer A containing 5mM MgCl$_2$ and different ATP concentrations. The curve was fitted to the Michaelis–Menten equation ($v = v_{max} * [ATP]/(K_m + [ATP])$). The Hill coefficient was determined from the slope of a plot of log ([ATP]) vs log ($v/(v_{max} - v)$). ATPase activity of $Pf$ FlaI, $Pf$ FlaI-NTD, $Pf$ FlaI-CTD and the stoichiometric mixture of $Pf$ FlaI-NTD and $Pf$ FlaI-CTD was determined by incubating the proteins at a concentration of 12.5 μg/ml in buffer A containing 5 mM MgCl$_2$ at 70 °C.

## Microscale thermophoresis

Microscale thermophoresis was performed as described previously (*Chaudhury et al., 2016*). To determine the binding affinity of CTD and NTD of $Pf$ FlaI, 28 nM labeled $Pf$ FlaH in buffer A containing 0.05% (v/v) Tween-20 was titrated with increasing concentrations (1.3 nM–22.5 μM) of $Pf$ -CTD and (5.6 nM–11.6 μM) $Pf$ -NTD of $Pf$ FlaI on a Nano Temper Monolith NT.115 Pico instrument. The data were fitted as described previously (*Chaudhury et al., 2016*).

## RESULTS

### Overproduction, purification and characterization of *P. furiosus* FlaI

To compare the crenarchaeal and euryarchaeal archaellum, we set out to biochemically characterize the archaellum of *Pyrococcus furiosus*. The genetic region that encodes the components of the archaellum of *P. furiosus* encodes the major archaellin (*flaB0*), followed by two minor archaellins (*flaB1, flaB2*) and the *flaCDFGHIJ* genes (*Robb et al., 2001*; *Näther et al., 2006*; *Näther-Schindler et al., 2014*). Here, we focus on the FlaI and FlaH proteins. Similar to *Sa*FlaI, *Pyrococcus furiosus* FlaI (*Pf* FlaI) contains both an N-terminal and a C-terminal domain, which show respectively 36% and 57% identity with *Sa*FlaI. *Pf* FlaI was previously overexpressed and purified (*Chaudhury et al., 2016*), and is here characterized further. Analysis of the oligomeric state of *Pf* FlaI using size exclusion chromatography showed that *Pf* FlaI eluted at a position corresponding to a hexamer (Figs. 2A and 2B). Thus, in this respect, *Pf* FlaI differs from *Sa*FlaI, which after purification elutes as a monomer and formed a hexamer in the presence of the non-hydrolysable ATP analog adenylyl-imidodiphosphate (AMP-PNP) (*Ghosh et al., 2011*) (Figs. 2A and 2B). The Abs$_{260}$/Abs$_{280}$ ratio of 0.3 suggested that *Pf* FlaI was isolated in the nucleotide free form. Indeed, further purification steps, dialysis or ammonium sulphate precipitation did not result in a change in the Abs$_{260}$/Abs$_{280}$ ratio. To test whether *Pf* FlaI could bind ATP, titrations with the fluorescent ATP analog MANT-ATP were performed. Similar experiments were performed to determine the nucleotide binding affinity of *Sa*FlaI (*Ghosh et al., 2011*). For *Pf* FlaI, the fluorescence increased linearly upon addition of MANT-ATP, until a maximum was reached after which no further increase was observed. This saturation was dependent on the protein concentration, and when 20 nM, 100 nM or 5 μM protein *Pf* FlaI (Fig. 3A) was used, the maximum was reached for all three concentrations at 1/3 of the *Pf* FlaI concentration used, demonstrating that only two of the ATP binding sites in the hexamer are available for binding of MANT-ATP. Indeed, *Sa*FlaI crystallized as a hexameric

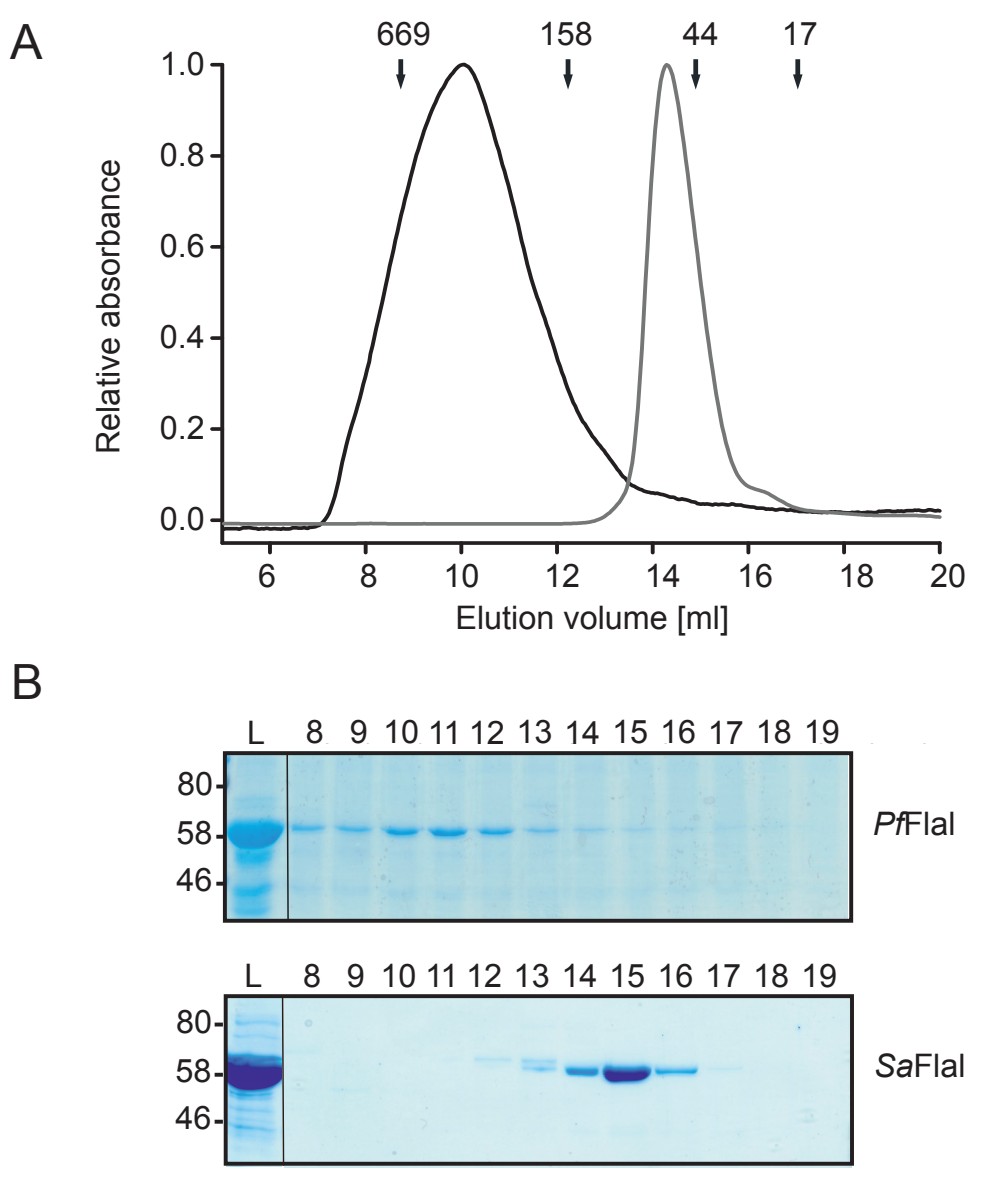

**Figure 2** *Pf* **FlaI forms a stable hexamer.** (A) Relative absorbance at 280 nm of size exclusion chromatography of *Pf* FlaI (64 kDa) shown in black line and *Sa*FlaI (59 kDa) is shown in grey line using Superdex 200 10/300GL column. Elution positions of molecular mass standards (kDa) were indicated with arrows. (B) Coomassie stained SDS-PAGE analysis of the different elution fractions.

ring with an intrinsic twofold symmetry resulting in three different conformations of the monomers (*Reindl et al., 2013*), suggesting that also for *Pf* FlaI such a twofold symmetry with three unique subunits occurs. To determine the affinity for ATP, bound MANT-ATP was competed with ATP resulting in an $IC_{50}$ of 260 nM at 20 °C (Fig. 3B). The hydrolysis of ATP by *Pf* FlaI was tested at different temperatures and at different pHs (Figs. 3C and 3D) and the highest activity was found at 70 °C and pH 8.0. Even though *P. furiosus* can live up to 103 °C, the ATPase activity *in vitro* decreased above 70 °C suggesting that at temperatures

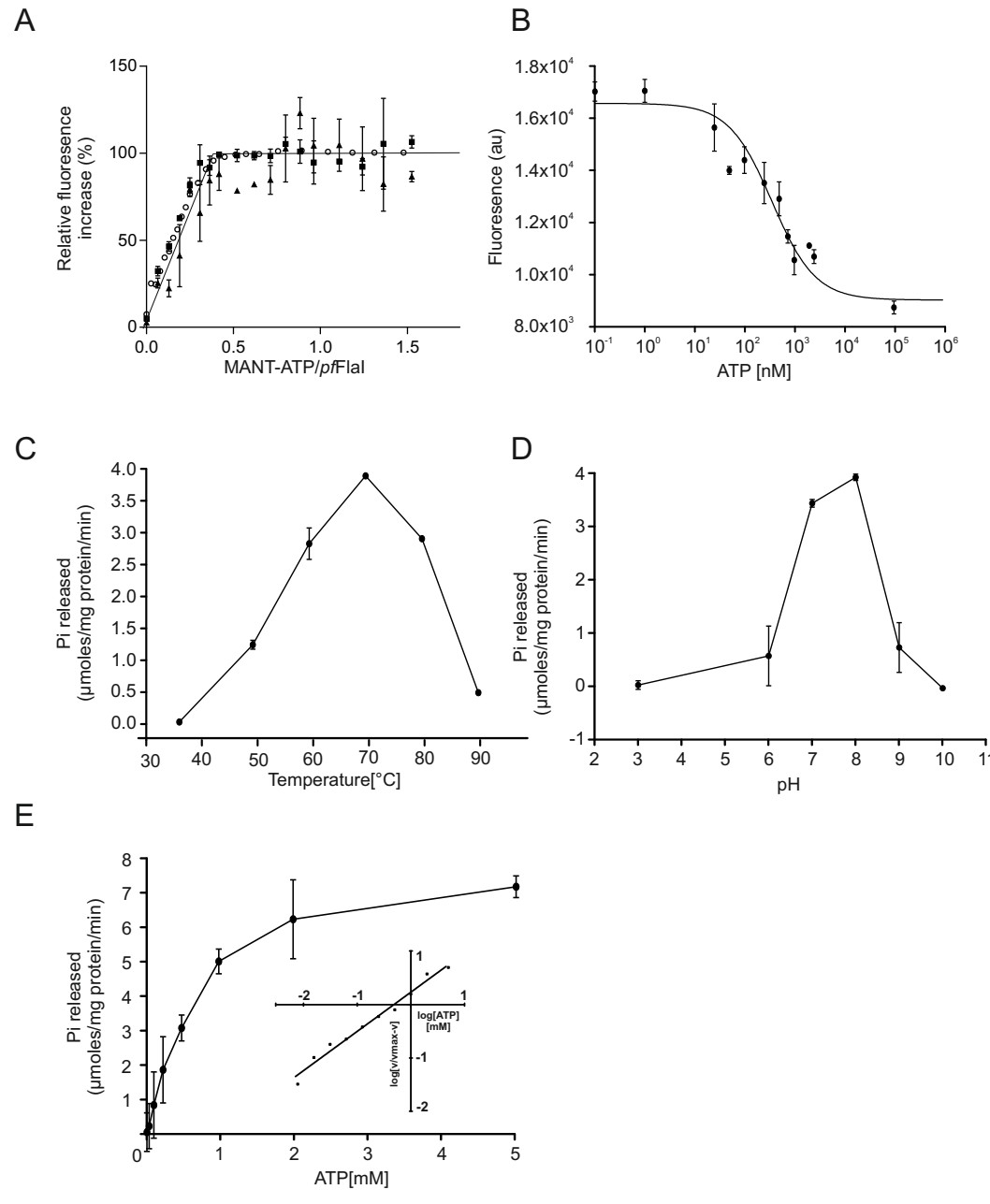

**Figure 3  ATP binding and hydrolysis of *Pf* FlaI.** (A) Fluorescence increase at increasing concentrations of MANT-ATP upon addition of 20 nM, 100 nM and 5 μM *Pf* FlaI. Lines depict the linear fits of the two observed phases. The lines cross at a MANT-ATP concentration of 1.7 μM. (B) Total fluorescence after addition of increasing amounts of ATP to a solution containing 20 nM *Pf* FlaI and 10 nM of MANT-ATP. The data were fitted with the Hill equation: $F = (F_{max} + (F_{min} - F_{max}) * [ATP]^n)/(IC_{50}^n + [ATP]^n)$ resulting in a best fit ($R^2 = 0.98$) with $IC_{50} = 260$ nM and $n = 0.67$. (C, D) ATP hydrolysis by 12.5 μg/ml *Pf* FlaI at different temperatures and at different pHs respectively. (E) ATPase activity of *Pf* FlaI at different ATP concentrations. The curve was fitted to the Michaelis-Menten equation ($V = V \max * [ATP]/(Km + [ATP])$), resulting in a $K_m$ of 580 nM. The inset shows the same data plotted according to the Hill equation (Hill coefficient = 0.9). Experiments were performed with at least two biological and three technical replicates. Error bars depict the standard error obtained from the technical replicates.

above 70 °C $Pf$ FlaI is unstable. At 70 °C and pH 8.0, ATP was hydrolyzed with a $v_{max}$ of 8 $\mu$moles mg$^{-1}$ min$^{-1}$ (*Chaudhury et al., 2016*) and a $K_m$ of 580 $\mu$M (Fig. 3E). ATP hydrolysis did not show cooperativity with increasing ATP concentrations (Hill coefficient = 0.9) (Fig. 3E, inset). The maximum activity observed for $Pf$ FlaI was 250-fold higher than the maximum activity (at pH 6.5 and 75 °C) observed for $Sa$FlaI (*Chaudhury et al., 2016*), and equals a turn-over of $\sim$500 ATP min$^{-1}$.

### The N-terminal domains of *Pf*FlaI interact with each other and with the C-terminal domains to stimulate ATP hydrolysis

The crystal structures of $Sa$FlaI revealed that, similar to other described T2SS and T4PS ATPases (*Robien et al., 2003*; *Yamagata & Tainer, 2007*; *Satyshur et al., 2007*; *Misic, Satyshur & Forest, 2010*; *Lu et al., 2013*; *McCallum et al., 2017*), $Sa$FlaI consists of a variable NTD and a CTD that binds and hydrolyses ATP (*Reindl et al., 2013*). These domains are connected via a short flexible linker (*Reindl et al., 2013*). The hexameric structure of $Sa$FlaI was only observed in the crystal structure after incubation with AMP-PNP. Since $Pf$ FlaI forms a much more stable hexamer, we set out to study the interactions between the NTDs and CTDs of $Pf$ FlaI.

Both the NTD ($Pf$ FlaI-NTD; 29 kDa) and the CTD ($Pf$ FlaI-CTD; 31 kDa) of $Pf$ FlaI were overexpressed in *E. coli*, purified, and were then analyzed on analytical size exclusion chromatography (Figs. 4A and 4B). The $Pf$ FlaI-CTD eluted as a monomer whereas, contrary to what was expected, the $Pf$ FlaI-NTD eluted not only as monomer but also as dimer and possibly a higher order oligomer. This demonstrates that the NTDs of $Pf$ FlaI interact with each other and might play a role in the formation of the hexameric ring. Since in the crystal structure of the $Sa$FlaI hexamer the NTDs interact with the CTDs of the neighboring subunit, it was tested whether also the $Pf$ FlaI-NTD and the $Pf$ FlaI-CTD interacted. Equimolar concentrations of $Pf$ FlaI-NTD and $Pf$ FlaI-CTD were mixed, analyzed on analytical size exclusion chromatography. This resulted in a co-elution of the $Pf$ FlaI-NTD and $Pf$ FlaI-CTD and a shift of especially the elution position of the $Pf$ FlaI-CTD, demonstrating an interaction between the $Pf$ FlaI-NTD and the $Pf$ FlaI-CTD. To test whether interaction between the $Pf$ FlaI-NTD and the $Pf$ FlaI-CTD influenced the ATPase activity, ATP hydrolysis of the single domains and of the mixed domains was determined (Fig. 4C). In these experiments, ATP hydrolysis was only observed when both the $Pf$ FlaI-NTD and $Pf$ FlaI-CTD were present, demonstrating that the $Pf$ FlaI-NTD can stimulate ATP hydrolysis by the $Pf$ FlaI-CTD.

### *Pf*FlaH stimulates the ATPase activity of *Pf*FlaI

We have overexpressed and purified *P. furiosus* FlaH ($Pf$ FlaH) and used microscale thermophoresis (MST) to show that nucleotide-bound $Pf$ FlaH bound to $Pf$ FlaI with a $K_D$ of 1 $\mu$M (*Chaudhury et al., 2016*). $Pf$ FlaH containing mutations in the Walker A (*Pf* FlaH K39A) and Walker B motifs (*Pf* FlaH D126N) had a strongly reduced affinity for nucleotides, and for $Pf$ FlaI, demonstrating that nucleotide binding to $Pf$ FlaH is important for its interaction with $Pf$ FlaI (*Chaudhury et al., 2016*). Here, we further investigated the influence of this interaction on the ATPase activity of $Pf$ FlaI. As observed, $Pf$ FlaI

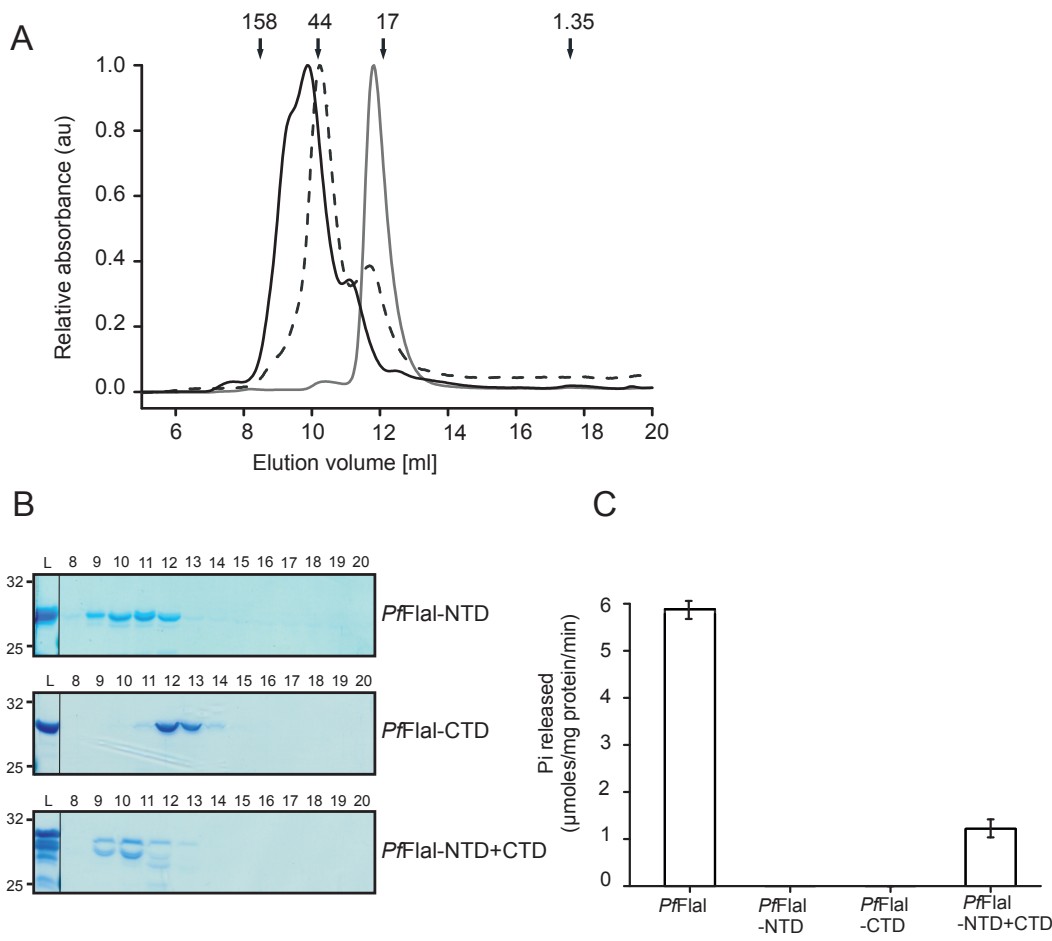

**Figure 4** **Analysis of the interaction between the N- and C-terminal domains of *Pf* FlaI.** (A) Relative absorbance at 280 nm of size exclusion chromatography at 280 nm of size exclusion chromatography of *Pf* FlaI-NTD (black line), *Pf* FlaI-CTD (grey line) or a stoichiometric mixture of *Pf* FlaI-NTD and *Pf* FlaI-CTD (dashed line) using Superdex 75 10/300GL column. Elution positions of molecular mass standards (kDa) are indicated with arrows. (B) SDS-PAGE analysis of the elution fractions described in A. (C) AT-Pase activity at 70 °C of the main elution fraction after size exclusion chromatography of *Pf* FlaI, *Pf* FlaI-NTD, *Pf* FlaI-CTD and the stoichiometric mixture of *Pf* FlaI-NTD and *Pf* FlaI-CTD at a concentration of 12.5 µg/ml. For A and B, a representative experiment is shown. For C, graph shows the average of two bi-ological replicates with two technical replicates.

hydrolyzed ATP, whereas for *Pf* FlaH, no ATPase activity could be observed (Fig. 5A, *Chaudhury et al., 2016*). Also, no ATP hydrolysis was observed for *Pf* FlaI with an E336A mutation in the Walker B motif, and for the *Pf* FlaHK39A and *Pf* FlaHD126N proteins. Addition of *Pf* FlaH to *Pf* FlaI stimulated the total ATPase activity. To test whether the stimulation of the ATPase is derived from an increase of the activity of *Pf* FlaH or of *Pf* FlaI, different combinations of mutants in *Pf* FlaH and *Pf* FlaI with WT proteins were tested (Fig. 5A). This demonstrated unequivocally that binding of nucleotide bound *Pf* FlaH stimulates the ATPase activity of *Pf* FlaI. To test the stoichiometry of this interaction, 1 µM *Pf* FlaI was incubated with increasing concentrations of *Pf* FlaH. A maximal stimulation

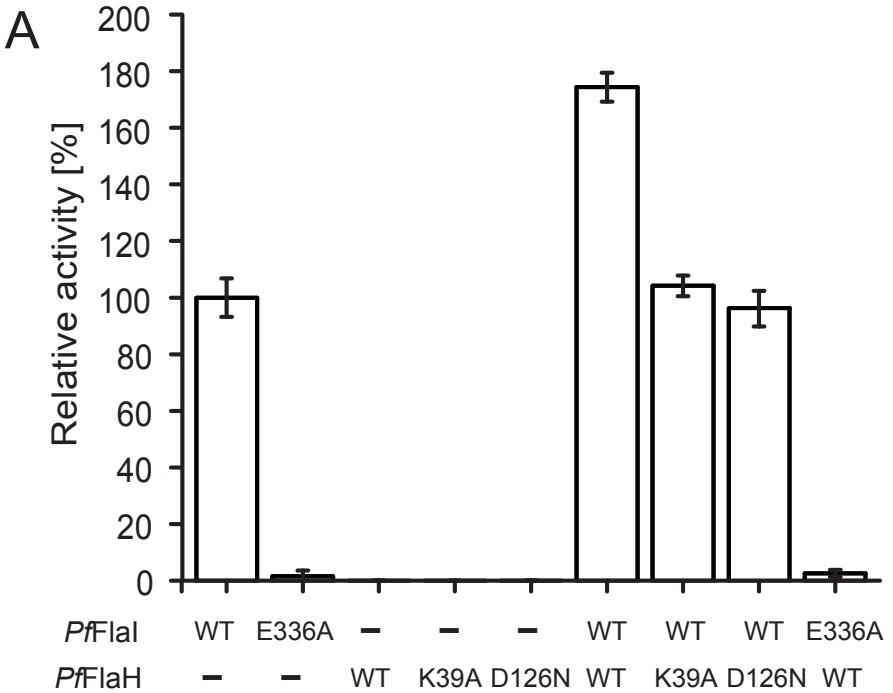

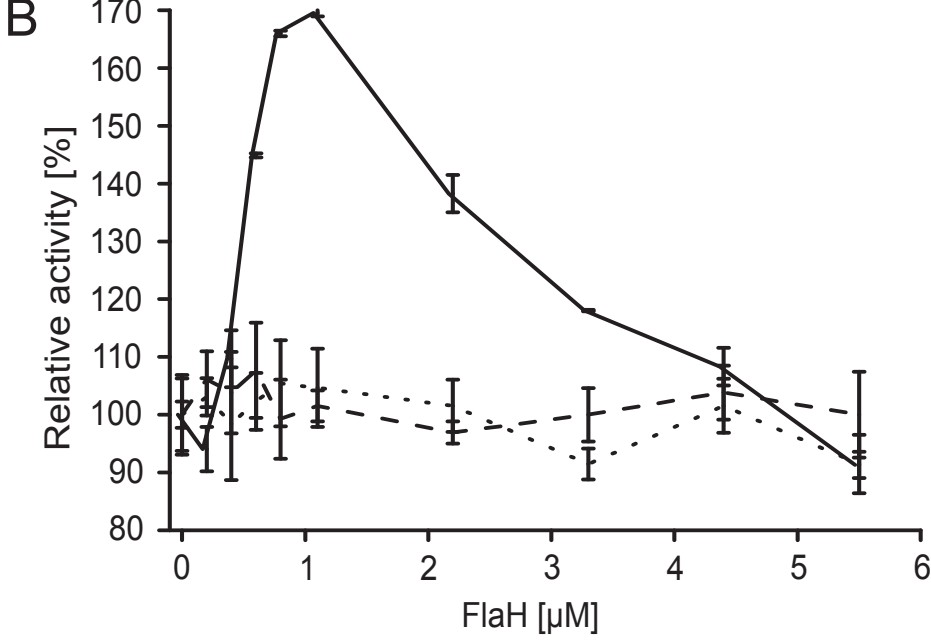

**Figure 5  Nucleotide bound *Pf* FlaH stimulates the ATPase activity of *Pf* FlaI.** (A) The ATPase activity at 70 °C was determined for *Pf* FlaI, *Pf* FlaH and these proteins with mutations in their respective Walker A *Pf* FlaH(K39A) and walker B motifs *Pf* FlaI(E336A), *Pf* FlaH(D126N) and of different combinations of these proteins. Proteins were added to a final concentration of 1 μM. (B) The ATPase activity at 70 °C was determined for 1 μM *Pf* FlaI and increasing amounts of *Pf* FlaH, *Pf* FlaH(K39A) and *Pf* FlaH(D126N) shown in solid line, dashed line and dotted line respectively. The graphs show the average of two biological replicates with two technical replicates.

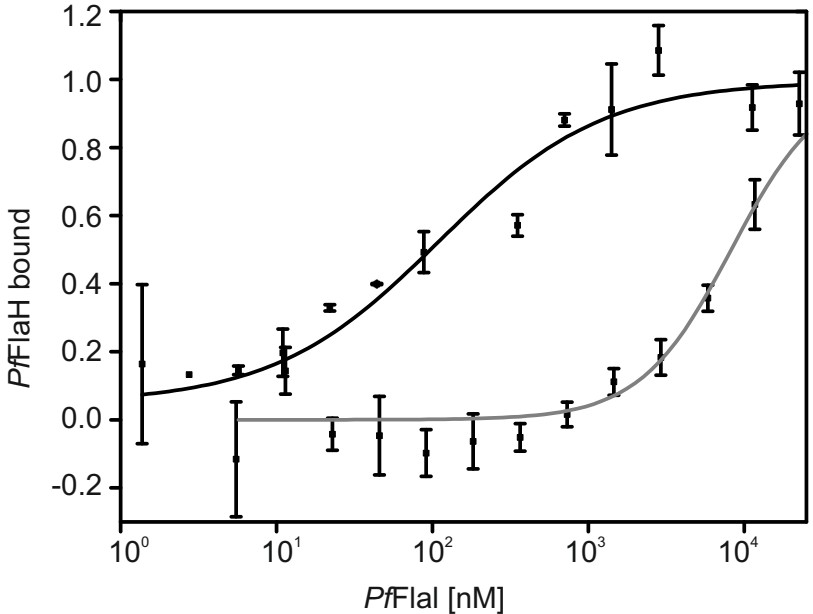

**Figure 6** *Pf* **FlaH interacts with higher affinity with *Pf* FlaI-CTD than *Pf* FlaI-NTD.** The binding of 28 nM fluorescently labeled Pf FlaH to increasing concentrations of 1.4 nM–22.5 µM for *Pf* FlaI-CTD (black) and 5.7 nM–11.7 µM for *Pf* FlaI-NTD (grey) was studied by microscale thermophoresis. Binding is depicted as the fraction bound and binding curves were fitted to the Hill equation. Curves were obtained from at least two independent experiments.

was observed when *Pf* FlaH was present in stoichiometric amounts to *Pf* FlaI (Fig. 5B). The stimulation decreased when further increasing amounts of *Pf* FlaH were added. Thus *Pf* FlaI and *Pf* FlaH interact in a 1:1 stoichiometry.

### FlaH interacts at the C-terminal domain of FlaI

Nucleotide-bound *Pf* FlaH can interact with *Pf* FlaI with an affinity of 1 µM (*Chaudhury et al., 2016*). To test whether FlaH interacts with the NTD or CTD of FlaI, interaction assays using microscale thermophoresis were performed using *Pf* FlaH and the *P. furiosus* NTD and CTD of FlaI (Fig. 6). The *Pf* FlaI-CTD and the *Pf* FlaI-NTD interacted with *Pf* FlaH with affinities of 100 nM and >8 µM, respectively. Thus, it was concluded that *Pf* FlaH interacts specifically with the *Pf* FlaI-CTD. This interaction occurs with a 10-fold higher affinity than the interaction between full length hexameric *Pf* FlaI and *Pf* FlaH, showing that *Pf* FlaH can interact in different manners with the hexameric *Pf* FlaI and the monomeric *Pf* FlaI-CTD.

## DISCUSSION

Many archaeal cell surface structures are homologous to bacterial T4PSs, which function in cell attachment to surfaces, DNA transport, biofilm formation and motility (*Jarrell et al., 2013*; *Makarova, Koonin & Albers, 2016*; *Chaudhury, Quax & Albers, 2018*). Similar to the T2SSs and T4PSs, most of these systems possess a pilin protein, a prepilin peptidase which

cleaves at the N-terminus of the prepilin, an ATPase of the superfamily of traffic ATPases, and a membrane platform protein (*Peabody, 2003*; *Nishida & Chen, 2004*). The traffic ATPase was suggested to interact with the membrane platform protein for several systems (*Reindl et al., 2013*; *Banerjee et al., 2013*; *Bischof et al., 2016*; *Takhar et al., 2013*), and it has been proposed that interactions between the processed pilin, the traffic ATPase protein and the platform protein drive the assembly of the pilus (*Chang et al., 2016*; *McCallum et al., 2017*). FlaI, the traffic ATPase of the archaellum differs from the other traffic ATPases in the fact that it not only energizes the assembly of the archaella, but also should drive its rotation. Comparison of the different crystal structures of traffic ATPases showed many different conformations of the NTD relative to the CTD, but in general, binding of ATP results in large domain movements bringing the NTD and CTD closer together (*Reindl et al., 2013*; *Lu et al., 2013*; *Satyshur et al., 2007*). Comparison of the conformations in *Sa*FlaI and *Vc* GspE, the traffic ATPase of the T2SS of *Vibrio cholerae*, showed that in *Vc* GspE, ATP hydrolysis results in an up and down movement of the domains, whereas, *Sa*FlaI shows a more rotating movement of the domains, possibly explaining the differences between *Sa*FlaI and other traffic ATPases (*Reindl et al., 2013*). The rotating movement of the *Sa*FlaI hexamer is the result of three different alternating conformations which mostly differ in the position of the NTD relative to the CTD.

Here the euryarchaeal *Pf* FlaI was characterized, and compared to the well characterized crenarchaeal *Sa*FlaI. Firstly, it was observed that *Pf* FlaI forms a stable hexamer whereas *Sa*FlaI was a monomer in solution. The *Sa*FlaI hexamer was observed only after incubation with a non-hydrolyzable ATP analogue and at high protein concentrations in the crystal structure. In the ADP-bound *Sa*FlaI crystal structure, three different alternating conformations were observed, which all contained a nucleotide. ATP binding assays with *Pf* FlaI showed that only two of the six positions in the *Pf* FlaI hexamer were accessible for MANT-ATP, suggesting that *Pf* FlaI also contains alternating conformations, and that four of the six subunits of the hexamer are not accessible to fluorescently labelled nucleotides. ATP hydrolysis by *Sa*FlaI was highly co-operative, whereas ATP hydrolysis by *Pf* FlaI did not show any cooperativity, suggesting that cooperatively occurs during the assembly of the *Sa*FlaI hexamer, while no cooperativity occurs in the assembled *Pf* FlaI hexamer. Indeed, the *Pf* FlaI hexamer is, contrary to *Sa*FlaI, formed in an ATP independent manner. This suggest that *Pf* FlaI and *Sa*FlaI might differ structurally. Another suggestion that *Pf* FlaI and *Sa*FlaI differ in their structures comes from the experiments with the isolated NTD and CTD. Whereas in the different *Sa*FlaI crystal structures obtained, interactions are found between the CTDs and between the CTDs and the NTD, no strong interactions were observed between the NTDs. Indeed, *Sa*FlaI lacking the NTD still exhibits 75% of the ATPase activity compared to the full-length FlaI (*Reindl et al., 2013*), suggesting that the NTD plays no important role in oligomerization. In *Pf* FlaI however, deletion of the NTD strongly reduces or abolishes oligomerization and ATPase activity. The activity and oligomerization can be partly recovered by the addition of the isolated NTD, suggesting that the NTD for *Pf* FlaI also plays an intrinsic role in oligomerization. Based on the crystal structure of *Sa*FlaI, it could be expected that this is caused by the stabilization of the interaction between two CTDs by an NTD, but our observation that the isolated NTDs

of *Pf* FlaI also oligomerize suggests that interactions between NTDs might also play a role in the oligomerization and activity of *Pf* FlaI. Strong interactions between NTDs of traffic ATPases, have currently only been observed for the HP0525 traffic ATPase of the *Helicobacter pylori* type IV secretion system (*Yeo et al., 2000*).

Next to the characterization *Pf* FlaI, the interaction of *Pf* FlaI with *Pf* FlaH was further analyzed. Binding of nucleotide bound *Pf* FlaH to *Pf* FlaI stimulated its ATPase activity twofold, further demonstrating the importance of this interaction. Maximum stimulation was found at a 1:1 stoichometry, suggesting that hexameric *Pf* FlaH interacts with hexameric *Pf* FlaI. Previously, it was demonstrated that both *Sa*FlaI and *Sa*FlaH interact with *Sa*FlaX (*Banerjee et al., 2013*). *In vitro* assembly of the C-terminal domain of *Sa*FlaX resulted in ring-like structures with 15- to 23- fold symmetry with widely different diameters (*Banerjee et al., 2012*). After incubation with *Sa*FlaH, monomeric *Sa*FlaH particles were observed inside these *Sa*FlaX rings (*Chaudhury et al., 2016*). The amount of FlaH bound inside the rings varied with the size of the ring, but in the most occurring rings with a 20-fold symmetry, 9–10 *Sa*FlaH monomers could be observed (*Chaudhury et al., 2016*). The size of the *in vivo* FlaX ring in the *S. acidocaldarius* archaellum complex is currently still unknown. However, since *Sa*FlaX also interacts with *Sa*FlaI, and a FlaI hexamer interacts with hexameric FlaH, it seems likely that the motor complex in *S. acidocaldarius* consists of hexameric *Sa*FlaI, bound to hexameric *Sa*FlaH, surrounded by a *Sa*FlaX ring. Like all euryarchaea, *P. furiosus* does not encode a FlaX homolog, but encodes the FlaC and FlaD proteins. However, it is currently not known whether they also form a ring-like structure and whether possible interactions with FlaI or FlaH exist. Remarkably, *P. furiosus* does not encode a chemotaxis system (*Maeder et al., 1999*) and thus the FlaCDE proteins should not be related to the CheY signal transduction cascade.

It was proposed that the NTD of FlaI interacts with FlaJ (*Reindl et al., 2013*; *Banerjee et al., 2013*), making it likely that the CTD domain of FlaI would interact with FlaH, and we found that *Pf* FlaH interacts with the *Pf* FlaI-CTD. This interaction occurs with a 10-fold higher affinity than the interaction between full length *Pf* FlaI and *Pf* FlaH. This suggest that the affinity of the interaction between FlaH and FlaI might be modulated by factors that increase or decrease the accessibility of the CTD of FlaI to FlaH, and this might facilitate a switch between assembly and rotation of the archaellum.

## CONCLUSIONS

Our results showed that FlaI of *P. furiosus* differs significantly from the extensively studied FlaI of *S. acidocaldarius*. Contrary to FlaI of *S. acidocaldarius*, which only forms a hexamer after binding of nucleotides, FlaI of *P. furiosus* forms a stable hexamer in a nucleotide independent manner. The presence of the stable hexamer allowed us to study nucleotide binding to the hexamer. This showed that only 2 of the 6 ATP binding sites were available for binding of the fluorescent ATP-analog MANT-ATP, suggesting a twofold symmetry in the hexamer and further suggesting that individual proteins in the hexamer alternate between the empty, ATP and ADP bound states. We also identified strong interactions between the N-terminal domains *S. acidocaldarius* FlaI not identified before for *S. acidocaldarius* FlaI.

These interactions played a role in oligomerization and activity, suggesting a conformational state of the hexamer not observed previously. We further showed that interaction between FlaI and FlaH stimulates the ATPase activity of FlaI. This occurs optimally at a 1:1 stoichiometry, suggesting that a FlaI hexamer can interact with six FlaH proteins or with a FlaH hexamer. Further interaction studies showed that FlaH interacts with the C-terminal domain of *Pf* FlaI.

## ACKNOWLEDGEMENTS

We thank Prof. Carola Hunte for giving access to Nanotemper Monolith NT.115 Pico instrument and Dr. Rashmi Kumariya for providing the data of the *Sa*FlaI gel filtration.

### Funding

Paushali Chaudhury and Sonja-Verena Albers were supported by a starting grant from the European Reserach Council (Nr. 311523, Archaellum). The funders had no role in study design, data collection and analysis, decision to publish, or preparation of the manuscript.

### Grant Disclosures

The following grant information was disclosed by the authors:
European Reserach Council: Nr. 311523.

### Competing Interests

Sonja-Verena Albers is an Academic Editor for PeerJ.

### Author Contributions

- Paushali Chaudhury conceived and designed the experiments, performed the experiments, analyzed the data, prepared figures and/or tables, authored or reviewed drafts of the paper, approved the final draft.
- Chris van der Does conceived and designed the experiments, analyzed the data, prepared figures and/or tables, authored or reviewed drafts of the paper, approved the final draft.
- Sonja-Verena Albers conceived and designed the experiments, analyzed the data, contributed reagents/materials/analysis tools, authored or reviewed drafts of the paper, approved the final draft.

### Data Availability

 The raw data are provided in a Supplemental File.

### Supplemental Information

Supplemental information for this article can be found online at http://dx.doi.org/10.7717/peerj.4984#supplemental-information.

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
