# Peer review of "Characterization of the ATPase FlaI of the motor complex of the Pyrococcus furiosus archaellum and its interactions between the ATP-binding protein FlaH"

_PeerJ, doi:10.7717/peerj.4984_

## Round 0.1 · original submission · Minor Revisions

The reviews are positive, however, there are comments you need to address before this manuscript can be accepted.

Reviewer 1 ·

Basic reporting

This group has been working on Archaellar flagella for years and in this paper they analysed the motor components FlaH and FlaI. Authors have already established a method to overproduce and purify these proteins and examined interaction between the two. Data presented here to show the direct interaction and effects on ATPase activity are convincing, and the description in the text is well written.
I have some questions below:

1. Fig.1A is not necessary to understand research backgrounds, because authors have not discussed any details of the atomic structures.
Instead, it will be nice to give a cartoon figure to show the relative locations of FlaH, FlaI, and FlaX at the base of archaellum.
Fig.1B is also not necessary in the present form. How about adding Walker motifs in the figure?


2. There is redundant usage of two words: previously and remarkably. Authors can write text (especially Discussion) without using them.

3. I do not see a logical connection between the last sentence (275-277) and the preceeding sentences.

4. Place a space in "aFlaH"(364).

Experimental design

Experimental design is conventional and easy to understand.

Validity of the findings

Not a big discovery but important to know to elucidate the mystrious motor function of Archaeal flagella.

Additional comments

I do not find any reasons to reject this paper.

·

Basic reporting

The manuscript describes the use of various methods for characterization of the Pyrococcus furiosus ATPase FlaI and its interactions with ATP-binding protein FlaHs and is a logical continuation of the experiments with the same objects described earlier (Chaudhury et al., Mol Microbiol (2016) 99: 674). The publication is fully justified. I have only minor remarks and recommendations to improve the manuscript.

Experimental design

The experimental design is clearly defined and performed at a high technical standard. The methods used are adequate and well described in detail.

Validity of the findings

The data obtained are novel and the presented information is valuable. The results are adequately documented with tables and figures.

Additional comments

The paper of Chaudhury et al. “Characterization of the interactions between the ATP-binding protein FlaH and the ATPase FlaI of the motor complex of the Pyrococcus furiosus archaellum” deals with an interesting problem of protein-protein interaction providing formation and function of the archaellum. This paper contains original experimental results and can be recommended for publication after certain minor changes. Some errors found in the text relate mainly to the correct arrangement of the literature cited.

Minor points.

1. Most of the article describes experiments on the characterization of the FlaI itself, which is not reflected in the title. It would be, the next title is more comprehensive “Characterization of the ATPase FlaI of the motor complex of the Pyrococcus furiosus archaellum and its interactions between the ATP-binding protein FlaH”.
2. Line 17. There should be "P. furiosus" instead of "P.fusiosus".
3. Line 20-21. It is indicated that Interactions between the N-terminal domains of FlaI were not observed previously/ But in “Discussion” (line 326-328 the authors claim that similar interaction was observed for traffic ATPase of the Helicobacter pylori. It will be more correct to say that such interactions were not observed for the crenarchaeal FlaI.
4. Lines 23-24. It should be add that a 1:1 stoichiometry suggests that hexameric PfFlaI interacts with hexameric PfFlaH.
5. Lines 47-48. Three references must be enclosed in one parenthesis.
6. Lines 54-56. It would be more correct to say about the "three unique subunit conformations".
7. Lines 61-63. 5 references must be enclosed in one parenthesis. Reference (Kenneth A. Satyshur…) should be presented as required.
8. Lines 102-105. Is there any information about the presence of homologues of the FlaCDE or FlaX in thaumarchaeota and nanoarchaeota?
9. Line 110. It would be more correct to say about the "the archaellar basal body".
10. Line 117. There should be "T. kodakaraensis" instead of "T. kodakaensis”.
11. Line 174. “Hill” should be written with a capital letter.
12. Lines 189-190. Three references must be enclosed in one parenthesis.
13. Lines 212-217. The Authors present the data on the PfFlaI ATPase activity at different temperatures. Are there any similar data for the SaFlaI? If yes, they must be compared.
14. Lines 224-226. See comment #7.
15. Lines 280-281; 284; 285-286; 288; 293-294). 47-48. The references should be presented as required.
16. In the “Discussion” (lines 333-343) the Authors discuss in detail the interaction of the SaFlaH and SaFlaI with SaFlaX, but do not make any assumptions about the interaction of the PfFlaH and PfFlaI with FlaX analogues, which may be FlaCDE. The P. furiosus does not have a chemotaxis system, and in this case the corresponding proteins should not be related to the CheY signal transduction cascade, similar to what the Authors write in the introduction (106-109).
17. Line 370. Apparently, "and" should stand between “gel filtration” and “SDS-PAGE”.
18. Lines 406-406. The name of the journal (Mol Microbiol, 99: 674-85) is not specified.
19. Lines 416, 423, 446, 468. Names of the organisms must be indicated in italics.
20. Lines 437-440. Reference (Kenneth A. Satyshur…) should be presented as required.
21. Line 480. The journal name should be written with a capital letter.

Reviewer 3 ·

Basic reporting

The article of Chaudhury et al. describes a focused and clearcut biochemical analysis of cytoplasmic archaeallum components from Pyrococcus furiosus. The authors describe their goals and their focus on this less studied model, which appears to offer advantages for biochemical analysis. The paper is clearly and succinctly written, while providing sufficient information in the Introduction and Discussion that place their study in a broader context.
The authors cite most of the recent and relevant literature.
I have two minor comments:

1. Line 210. It is not clear what the authors mean by "three unique subunits". Presumably there are three different conformations of monomers in the crystal, but this should be better explained.

2. Multiple references in the text and the list are incomplete or need reformatting. The authors should also group their references, when successive, together in a single pair of brackets. Examples include lines 47, 62, 224 and all references in the Discussion. Several references have parts missing or are misformatted in the list (e.g. lines 394, 404, 406, 421, 433, 437, 475, 480).

Experimental design

The authors describe purification and biochemical analysis of FlaI, the hexameric ATPase and the main motor of the system, and FlaH, an ATP-binding protein of the RecA family. The quality of this work meets the high standards and is supported by quantitative approach.


The authors show that Pyrococcus FlaI can be purified to homogeneity and behaves as a hexamer, even in the absence of ATP or analogues. unlike Sulfolobus FlaI used as a control. The authors further show that Pf FlaI binds ATP in vitro at a 3:1 molar ratio, suggesting that only two ATP binding sites are accessible in the hexamer and leading the authors to deduce a two-fold symmetry of the hexamer. Using the purified FlaI Pf the authors analyze the kinetic constants, optimum temperature and pH for its ATPase activity.

The two domains of FliI were produced separately and their multimeric state was tested by SEC. These experiments show that the N-terminal domain alone behaves as a hexamer whereas the C-terminal domain was monomeric. When the two domains were mixed, the C-domain co-eluted with the N-domain multimer, and this interaction was required for reconstitution of ATPase activity.
Next, the ATP binding protein FlaH was purified and its ability to stimulate FlaI ATPase activity was demonstrated in vitro. This stimulation required intact walker A and walker B boxes of FlaH. When FlaH was added to FlaI variant defective for ATPase activity, no ATP hydrolysis was detected, demonstrating that FlaH is unable to hydrolyse ATP, even in the presence of FlaI.

Finally, using microscale thermophoresis, the authors determined FliH binding affinities of to the N- and C-terminal domains of FlaI. These experiments showed higher affinity of FliH to the C-terminal domain, which was even higher than that observed for the FlaI hexamer. The C-terminal binding and 1:1 stoichiometry is consistent with positioning of FlaH hexamer underneath the FlaI hexamer at the base of the archaellum.

Validity of the findings

Overall, the experiments are well executed and support the conclusions. Proper controls are included in the experiments and statistical analysis is applied.

The Discussion highlights differences between Sulfolobus and Pyrococcus FlaI and the role of N-domain oligomerization in ATPase activity and in the cooperativity of the process. The results are placed in the context of recent structural data obtained by crystallography and cryo-tomography in different species.

Additional comments

None

---

## Round 0.2 · accepted · Accept

All comments were addressed.

# ·

Basic reporting

All my comments have been taken into account. The resulting manuscript has become more perfect. It meets the PeerJ criteria and should be accepted.

Experimental design

no comment

Validity of the findings

no comment

Additional comments

All my comments have been taken into account. The resulting manuscript has become more perfect.